# Arachidonoylcholine and Other Unsaturated Long-Chain Acylcholines Are Endogenous Modulators of the Acetylcholine Signaling System

**DOI:** 10.3390/biom10020283

**Published:** 2020-02-12

**Authors:** Mikhail G. Akimov, Denis S. Kudryavtsev, Elena V. Kryukova, Elena V. Fomina-Ageeva, Stanislav S. Zakharov, Natalia M. Gretskaya, Galina N. Zinchenko, Igor V. Serkov, Galina F. Makhaeva, Natalia P. Boltneva, Nadezhda V. Kovaleva, Olga G. Serebryakova, Sofya V. Lushchekina, Victor A. Palikov, Yulia Palikova, Igor A. Dyachenko, Igor E. Kasheverov, Victor I. Tsetlin, Vladimir V. Bezuglov

**Affiliations:** 1Department of molecular neuroimmune signaling, Shemyakin-Ovchinnikov Institute of Bioorganic Chemistry, Russian Academy of Sciences, Moscow 117997, Russia; kudryavtsevden@gmail.com (D.S.K.); evkr@mail.ru (E.V.K.); evfa57@gmail.com (E.V.F.-A.); xelor.craiz@gmail.com (S.S.Z.); natalia.gretskaya@gmail.com (N.M.G.); zgn55@yandex.ru (G.N.Z.); vpalikov@bibch.ru (V.A.P.); yuliyapalikova@bibch.ru (Y.P.); dyachenko@bibch.ru (I.A.D.); shak_ever@yahoo.com (I.E.K.); victortsetlin3f@gmail.com (V.I.T.); vvbez2013@yandex.ru (V.V.B.); 2Department medicinal and biological chemistry, Institute of Physiologically Active Compounds, Russian Academy of Sciences, Chernogolovka 142432, Moscow Region, Russia; serkoviv@mail.ru (I.V.S.); galina.makhaeva@gmail.com (G.F.M.); boltneva@ipac.ac.ru (N.P.B.); kovalevanv@ipac.ac.ru (N.V.K.); sog@ipac.ac.ru (O.G.S.); sofya.lushchekina@gmail.com (S.V.L.); 3Department of electrophysics of organic materials and nanostructures, Emanuel Institute of Biochemical Physics, Russian Academy of Sciences, Moscow 119334, Russia; 4Institute of Molecular Medicine, Sechenov First Moscow State Medical University, Moscow 119991, Russia

**Keywords:** nAChR, arachidonoylcholine, oleoylcholine, acylcholines, acetylcholinesterase

## Abstract

Cholines acylated with unsaturated fatty acids are a recently discovered family of endogenous lipids. However, the data on the biological activity of acylcholines remain very limited. We hypothesized that acylcholines containing residues of arachidonic (AA-CHOL), oleic (Ol-CHOL), linoleic (Ln-CHOL), and docosahexaenoic (DHA-CHOL) acids act as modulators of the acetylcholine signaling system. In the radioligand binding assay, acylcholines showed inhibition in the micromolar range of both α7 neuronal nAChR overexpressed in GH4C1 cells and muscle type nAChR from *Torpedo californica*, as well as *Lymnaea*
*stagnalis* acetylcholine binding protein. Functional response was checked in two cell lines endogenously expressing α7 nAChR. In SH-SY5Y cells, these compounds did not induce Ca^2+^ rise, but inhibited the acetylcholine-evoked Ca^2+^ rise with IC_50_ 9 to 12 μM. In the A549 lung cancer cells, where α7 nAChR activation stimulates proliferation, Ol-CHOL, Ln-CHOL, and AA-CHOL dose-dependently decreased cell viability by up to 45%. AA-CHOL inhibited human erythrocyte acetylcholinesterase (AChE) and horse serum butyrylcholinesterase (BChE) by a mixed type mechanism with *K_i_*= 16.7 ± 1.5 μM and *αK_i_*= 51.4 ± 4.1 μM for AChE and *K_i_*= 70.5 ± 6.3 μM and *αK_i_*= 214 ± 17 μM for BChE, being a weak substrate of the last enzyme only, agrees with molecular docking results. Thus, long-chain unsaturated acylcholines could be viewed as endogenous modulators of the acetylcholine signaling system.

## 1. Introduction

A system for the synthesis, reception, transport, and hydrolysis of acetylcholine (ACh) is present in most types of human cells [1]. In addition to the functions of a neurotransmitter, ACh regulates many processes, such as proliferation, differentiation, apoptosis, locomotor activity, angiogenesis, immune functions, secretion, organization of the cytoskeleton, and others [2]. Choline acetyltransferase, the key enzyme responsible for acetylcholine synthesis, is known to be unselective for the fatty acids that esterify choline. In rat brain, choline acetyltransferase exhibits the same affinity for butyryl-CoA and propionyl-CoA as for acetyl-CoA [3], leading to the production of propionylcholine and butyrylcholine in the presence of the respective precursor substances. Indeed, acetylcholine analogs with butyric, propionic, caproic, caprylic, and palmitic acids were identified in mammalian tissues [4,5,6]. As compared to ACh, these compounds possessed weaker muscarinic-like activity and significantly stronger activity nicotinic-like activity [4] in physiological response tests.

Due to a low production rate and a lower affinity of these atypical esters for the ACh receptors, they are considered to be modulators of cholinergic signaling due to the desensitization of the receptors against the dominant signal molecule, acetylcholine [6].

Recently, novel ACh analogs with the chain length of 18 to 22 carbon atoms and various degree of unsaturation of fatty acid residue were discovered. Normal human plasma and urine contain choline derivatives with arachidonic, docosahexaenoic, oleic, linoleic, linolenic, and palmitic fatty acid residues [7]. More importantly, the increased content of long-chain unsaturated acylcholines was found in several diseases. Thus, oleic, linoleic, and arachidonic acid derivatives of choline were identified in the tissues of blood vessels obtained from the operations of patients suffering from cardiovascular diseases (abdominal aortic aneurysm, stenotic carotid plaques, femoral stenotic plaques, and intimal thickening) [8]. Arachidonoylcholine (AA-CHOL) was found in the profile of metabolites in atherosclerotic plaques and the adjacent intima of the arteries [9]. Acylcholines, especially the unsaturated ones, were increased in blood samples obtained from patients with a high risk of pulmonary embolism (PE) as compared with samples of patients with an intermediate risk of PE [10]. In addition, the increased level of eicosapentaenoylcholine correlated with the level of 25-hydroxy vitamin D in a study on osteoporosis control in Hong Kong [11].

The described data lead to a hypothesis that acetylcholine analogs with unsaturated long-chain fatty acid residue could play a significant role in disease formation. However, the data on the activity and targets for the long-chain unsaturated acylcholines are limited to a sea urchin *Strongylocentrotus droebachiensis* and *S. purpuratus* embryo development model. Unfertilized sea urchin eggs express ACh receptors resembling the neuronal nicotinic ACh receptors (nAChR) [12]. AA-CHOL and docosahexaenoylcholine (DHA-CHOL) dose-dependently induced the larva immobilization and cell lysis, and noncompetitive nicotinic cholinergic antagonist QX-222 quenched this effect [13]. Thus, AA-CHOL acted as a nAChR agonist or a cholinomimetic. However, the sea urchin ACh receptor-like protein is still poorly characterized, and therefore more data in mammalian models are required to understand the role of these compounds.

Considering the above facts, we hypothesized that fatty acid analogues of acetylcholine can function in the body as modulators of the acetylcholine system. To verify this assumption, we synthesized choline fatty acid esters with chain lengths C18, C20, and C22, namely oleic, linoleic, arachidonic, and docosahexaenoic acids. In this paper, we focused on the nicotinic acetylcholine system. We studied the ability of choline esters to interact with neuronal and muscle type acetylcholine receptors, determined the functionality of this interaction using cell cultures, checked the possible physiological effect of AA-CHOL in vivo, and also found out whether such acylcholines are substrates of acetylcholine hydrolysis enzymes and/or their inhibitors.

As the result, we show for the first time that arachidonoylcholine and its unsaturated fatty acid analogs with the chain length of 18 and 22 carbon atoms are inhibitors of the neuronal and muscle-type nicotinic receptors and modest inhibitors of the acetylcholinesterase (AChE, EC 3.1.1.7) and butyrylcholinesterase (BChE, EC 3.1.1.8), and thus could act as endogenous modulators of the acetylcholine signaling system.

## 2. Materials and Methods

### 2.1. Reagents

l-Glutamine, fetal bovine serum, penicillin, streptomycin, amphotericin B, HEPES, Hanks’ salts solution, trypsin, DMEM, (4,5-dimethylthiazol-2-yl)-2,5-diphenyltetrazolium bromide (MTT) were from PanEco, Moscow, Russia. HEPES, KCl, CaCl_2_, MgCl_2_, DMSO, Triton X-100, d-glucose, non-essential amino acids, Hoechst 33258, human erythrocyte AChE, equine serum BChE, acetylthiocholine iodide, CHAPS, EDTA, dithiothreitol, PNU 120596, protease inhibitor cocktail, SCP0139, and 5,5′-dithio-bis-(2-nitrobenzoic acid) (DTNB) were from Sigma-Aldrich, St. Louis, MO USA. Arachidonic acid and Z-VAD-FMK were purchased from Cayman Europe, Hamburg, Germany. Ac-DEVD-AFC and Ac-LEHD-AFC were from Tocris Bioscience, Bristol, UK. Fluo-4AM and probenecid were from Thermo Fisher Scientific, Waltham, MA, USA.

### 2.2. Animals

Specific pathogen-free outbred ICR male mice (6 to 8 weeks old, weighing 29 to 33 g) were obtained from the Animal Breeding Facility of the Branch of Shemyakin-Ovchinnikov Institute of Bioorganic Chemistry of the Russian Academy of Sciences (Pushchino). The animals had been acclimatized for 2 weeks before experimental procedures and were kept in two-corridor barrier rooms under a controlled environment: temperature 20 to 24 °C, relative humidity 30% to 60%, 12 h light cycle. Animals were housed in standard polycarbonate cages Type 3 (820 cm^2^) on bedding (LIGNOCEL BK 8/15, JRS, Germany), with ad libitum access to feed (SSNIFF V1534-300, Spezialdiaeten, GmbH) and filtered tap water. Mouse cages were also supplied with material for environmental enrichment, Mouse House™ (Techniplast, Italy). The study was conducted in AAALAC (Association for Assessment and Accreditation of Laboratory Animal Care International) accredited facility in compliance with the standards of the Guide for Care and Use of Laboratory Animals (8th edition, Institute for Laboratory Animal Research). Animal treatment procedures were approved by the Institutional Animal Care and Use Committee (IACUC) of the Branch of the Shemyakin-Ovchinnikov Institute of Bioorganic Chemistry, Russian Academy of Sciences, the experimental protocol code is no. 700/19.

### 2.3. Chemical Synthesis

Dimethylaminoethanol esters of arachidonic (AA-DMAE), docosahexaenoic (DHA-DMAE), linoleic (Ln-DMAE), and oleic (Ol-DMAE) acids were obtained by the treatment of corresponding fatty acid chloride with β-*N*,*N*-dimethylaminoethanol as described in [13]. AA-CHOL, DHA-CHOL, Ol-CHOL, and Ln-CHOL were synthesized by treatment and AA-DMAE, DHA-DMAE, Ol-DMAE, and Ln-DMAE with methyl iodide as described in [13]. The compound purities were 95% to 97%.

### 2.4. Cell Culture

SH-SY5Y cells (ATCC CRL-2266) were maintained in the DMEM medium supplemented with 10% of fetal bovine serum, 4 mM of l-glutamine, and 1% of non-essential amino acids. The A549 cells (ATCC CCL-185) were maintained in DMEM medium supplemented with 2 mM of l-glutamine and 10% of fetal bovine serum. All cell medium contained 100 U/mL of penicillin, 100 μg/mL of streptomycin, and 2.5 μg/mL of amphotericin B. The cells were cultured in an atmosphere of 95% humidity and 5% CO_2_ at 37 °C. Cells were passaged every 3 days and continuously grown for no more than 40 passages. Attached cells were detached with 0.25% trypsin in 0.53 mM EDTA in Hanks’ salts. Cells were counted using a glass hemocytometer. The cells were routinely checked for mycoplasma contamination using Hoechst 33258 staining with microscopic detection.

### 2.5. Cytotoxicity Assay

For the analysis of cell death induction, the cells were plated in 96-well plates at a density of 1.5 × 10^4^ cells per well and grown overnight. The dilutions of test compounds prepared in DMSO and dissolved in the culture medium were added to the cells in triplicate for each concentration so that every well contained 100 μL of conditioned medium and 100 μL of fresh medium with the test substance and incubated for 18 h. The incubation time was chosen based on the most pronounced differences between the compounds tested. The final DMSO concentration was 0.5%. Negative (all cells alive) control cells were treated with 0.5% DMSO. Positive (all cells dead) control cells were treated with 3.6 μL of 50% Triton X-100 in ethanol per 200 μL of cell culture medium. Separate controls were without DMSO (no difference with the control 0.5% DMSO was found, data not shown). The effect of test substances on the cell viability was evaluated using the MTT test (based on the MTT dye reduction by mitochondria of living cells) [14].

### 2.6. Caspase Activation Assay

A549 cells were plated in 12-well plates at a density of 4 × 10^5^ cells per well and grown overnight. The dilutions of test compounds prepared in DMSO and dissolved in the culture medium were added to the cells in triplicate for each concentration so that every well contained 400 μL of conditioned medium and 400 μL of fresh medium with the test substance and incubated for 5 h. The final DMSO concentration was 0.5%. After the incubation, the medium was collected and centrifuged for 5 min at 2000*g*; the pelleted dead cells and the remained attached cells were lysed with 200 μL of the lysis buffer (250 mM HEPES, pH 7.4, 50 mM EDTA, 2.5% CHAPS, 125 mM dithiothreitol, and protease inhibitor cocktail) per well for 10 min at +4 °C. The lysates were centrifuged for 15 min at 18,000*g,* at room temperature, and the pellets were discarded. Then, 50 μL of each lysate was incubated with 50 μL of the lysis buffer containing 50 μM of the appropriate caspase fluorogenic substrate for 2 h at 37 °C without stirring, after which the fluorescence was determined using the Hidex Sense Beta Plus microplate reader (Turku, Finland), λ_ex_. = 355 nm and λ_em_. = 535 nm.

### 2.7. Apoptosis Assay

A549 cells were plated in 48-well plates at a density of 1.5 × 10^4^ cells per well and grown overnight. The dilutions of test compounds prepared in DMSO and dissolved in the culture medium were added to the cells in triplicate for each concentration so that every well contained 200 μL of conditioned medium and 200 μL of fresh medium with the test substance and incubated for 2 h. The final DMSO concentration was 0.5%. After the incubation, the cells were stained using a Apoptosis/Necrosis assay kit (Abcam, Cambridge, MA, USA) according to the manufacturer’s protocol and imaged using a Nikon Ti-S fluorescent microscope.

### 2.8. Calcium Mobilization Assay

Cells were cultured for two days at 37 °C on black 96-well plates. Immediately before the experiments, cells were incubated with 2 mM Fluo-4AM ester reagent and 1.25 mM probenecid (organic anion transporter inhibitor) for 1 h at room temperature. After the incubation, the cells were washed out with the extracellular solution (140 mM NaCl, 2 mM CaCl_2_, 2.8 mM KCl, 4 mM MgCl_2_, 20 mM HEPES, 10 mM glucose; pH 7.4). The last washout was supplied with 10 uM α7 nAChR positive allosteric modulator PNU 120596 and tested compound. The excitation of Fluo-4 achieved at 485 nm and fluorescence registered at 535 nm using Hidex Sense Beta Plus (Turku, Finland) multi-well plate fluorimeter. The calcium rise amplitude was measured from the base level of each well.

### 2.9. Radioligand Competition Assay of Acylcholines with Radioiodinated A-Bungarotoxin

For radioligand competition assay, the targets were membrane-bound *T. californica* nAChR (kindly provided by Prof. F. Hucho, Institute for Chemistry and Biochemistry, Freie Universität Berlin, Germany), human α7 nAChR stably expressed in the rat pituitary tumor-derived cell line GH4C1 (received from Eli Lilly and Company, London, UK), and AChBP from *L. stagnalis* (kindly provided by Prof. S. Luo, Key Laboratory for Marine Drugs of Haikou, Hainan University, China).

We used suspensions of membranes from *T. californica* electric organ (1.25 nM α-bungarotoxin binding sites), α7 nAChR-transfected GH4C1 cells (0.4 nM), or the heterologously expressed *L. stagnalis* acetylcholine binding protein (AChBP) (2.4 nM) in 50 μL of binding buffer (Tris-HCl, 20 mM; BSA, 1 mg/mL; pH 8.0). Ligands were added in various amounts and incubated with continuous shaking for 60 min at 20 °C. After that, ^125^I-α-Bgt (500 Ci/mmol) was added at a final concentration of 0.1 nM, and samples were further incubated for 5 min. The membranes or cell preparations were filtered through GF/C glass filters (Whatman, Maidstone, UK), washed, and bound radioactivity was measured using a Wallac 1470 Wizard Gamma Counter (PerkinElmer, Waltham, MA, USA). The AChBP samples were incubated with 10 μL of Ni^2+^-NTA-agarose, and suspensions were filtered, washed, and bound radioactivity was measured as described above. The nonspecific ^125^I-αBgt binding was determined in the presence of a 200-fold excess of α-cobratoxin.

### 2.10. Electrophysiology

Oocytes were removed from mature Xenopus frogs treated with 2 mg/mL type “I” collagenase (Gibco, Life Technologies Corp., NY, USA) dissolved in a Ca^2+^-free ND96 buffer (5 mM HEPES, 2 mM MgCl_2_, 2 mM KCl, and 96 mM NaCl; pH 7.5). After 2 to 4 h of collagenase treatment, oocytes were transferred to regular ND96 (5 mM HEPES, 2 mM MgCl_2_, 1.8 mM CaCl_2_, 2 mM KCl, and 96 mM NaCl; pH 7.5) and injected with 1 to 5 ng of mouse muscle nAChR α1, β1, δ, and ε subunit cDNA in pRBG4 vector; mouse α1, β3, and γ2 GABAAR subunits cDNA in PCI vector. Recordings were performed 24 to 72 h after injection. A Turbo TEC-03X amplifier (NPI electronic, Tamm, Germany) was used along with the WinWCP software.

### 2.11. Assessment of the Cholinesterases Hydrolysis of AA-CHOL

AChE and BChE hydrolysis of AA-CHOL was determined in 100 mM phosphate buffer, pH 7.5 at 25 °C, the AA-CHOL concentration was 1 mM, the incubation time was 40 min. The enzyme concentrations were 0.02 U/mL in the sample. The reverse phase HPLC method was used to estimate the hydrolysis of AA-CHOL. As a control, acetylthiocholine iodide (ATCh) hydrolysis was determined in the same conditions.

### 2.12. Esterase Inhibition Assay

AChE and BChE activities were determined by the colorimetric Ellman method (λ 412 nm) [15], with some minor modifications as described in detail in [16], using ATCh (1 mM) as substrate. The assay solution consisted of 0.1 M K/Na phosphate buffer pH 7.5, 25 °C, 0.33 mM DTNB, 0.02 unit/mL AChE or BChE. Reagent blanks consisted of reaction mixtures without substrate. The measurements were carried out on a FLUOStar OPTIMA (BMG Labtech, Germany) microplate reader. The compounds were dissolved in DMSO, and the incubation mixture contained 2% of this solvent. 

Preliminary evaluation of the inhibitory activity of the compounds was carried out by determining the degree of inhibition of enzymes at a compound concentration of 100 μM. For this, an enzyme sample was incubated with a test compound for 10 min, then the residual enzyme activity was determined and compared with the enzyme activity without a tested compound. To determine the IC_50_, an enzyme sample was incubated with the test compound for 10 min, then, the residual enzyme activity was determined. The concentration range of the test compound was 1 × 10^−8^ to 1 × 10^−3^ M. Origin 6.1 for Windows was used to determine IC_50_ values from plots of log (inhibitor concentration) vs. % (residual activity). Results were expressed as mean ± SEM (*n* = 11 experiments).

### 2.13. Kinetic Analysis of AChE and BChE Inhibition by AA-CHOL: Determination of Steady-State Inhibition Constants

To elucidate the inhibition mechanisms for the active compound AA-CHOL, the AChE and BChE residual activities were determined in the presence of two increasing concentrations of AA-CHOL and six decreasing concentrations of the substrate. The test compound was preincubated with the enzymes at 25 °C for 10 min, followed by the addition of ATCh. Parallel controls were made to find the rate of hydrolysis of the same concentrations of substrate in the solutions with no inhibitor. The kinetic parameters of substrate hydrolysis were determined. The measurements were carried out using a BioRad Benchmark Plus microplate spectrophotometer (France). Each experiment was performed in triplicate. The results were fitted into Lineweaver–Burk double-reciprocal kinetic plots of 1/V versus 1/[S] and the values of inhibition constants *K_i_* (competitive component) and *αK_i_*(noncompetitive component) were calculated using Origin 6.1 software for Windows.

### 2.14. In Vivo Myorelaxant Activity

Tested AA-CHOL solution was injected into the triceps muscles of the mouse forelimbs at a volume of 0.5 mL/kg per each limb or subcutaneously. For all animals, their forelimb grip strength was recorded before the substance administration with a 1027 grip strength meter (Columbus Instruments, Columbus, OH, USA). Their grip strength was measured 15, 30, 60,120, and 240 min after the AA-CHOL or saline administration.

### 2.15. Molecular Docking

Geometry of AA-CHOL was quantum-mechanically optimized with Gamess-US [17] software (B3LYP/6-31G*). The optimized structure was used with partial atomic charges derived from QM results according to the Löwdin scheme. The most suitable for docking of bulky ligands [18] X-ray structure of human AChE, co-crystallized with donepezil PDB ID 4EY7 [19] and an optimized X-ray structure of human BChE (PDB ID 1P0I [20,21]) were used. Molecular docking was performed with AutoDock 4.2.6 software [22]. The grid box for docking included the entire active site gorge of AChE (22.5 Å × 22.5 × 22.5 Å. grid box dimensions) and BChE (15 Å × 20.25 Å × 18 Å grid box dimensions) with a grid spacing of 0.375 Å. The main Lamarckian genetic algorithm (LGA) [23] parameters were 256 runs, 25 × 10^6^ evaluations, 27 × 10^4^ generations, and a population size of 3000.

Figures were prepared with PyMol (www.pymol.org).

### 2.16. Statistical Procedures

All experiments were conducted three times. GraphPad Prism 6.0 was used to plot the data and calculate EC_50_. The data were compared using ANOVA with the Holm-Sidak post-test; *p* ≤ 0.05 was considered a statistically significant difference.

The radioligand assay results were analyzed with ORIGIN 8.0 (OriginLab Corporation, Northampton, MA, USA) fitting to a one-site dose-response curve by the equation: % response  =  100/{1  +  ([toxin]/IC_50_)*^p^*}, where IC_50_ is the concentration at which 50% of the binding sites are inhibited, and *p* is the Hill coefficient. Data in the radioligand assay are presented as mean with 95% confidence interval (CI).

## 3. Results

### 3.1. Chemical Synthesis Of Acylcholines

Choline esters of arachidonic (AA-CHOL), oleic (Ol-CHOL), linoleic (Ln-CHOL), and docosahexaenoic (DHA-CHOL) fatty acids (Figure 1) were synthesized via the dimethylaminoethanol ester intermediates according to the following scheme (Scheme 1), as was previously described [13]. The intermediate compounds were purified using low-pressure column silica chromatography, while target acylcholines were recovered after quaternization in a pure form by evaporation of reaction mixture. The structures were validated using ^1^H-NMR. The reaction yields were about 50%, and the obtained compound purities were 95% to 97% as was tested by reversed phase HPLC.

### 3.2. Acylcholines Interact with the A-Bungarotoxin Binding Site on nAChR

Using a radioligand assay, we studied acylcholines for their ability to compete with ^125^I-αBgt binding to the orthosteric sites of the nAChRs. At first, we tested the binding of acylcholines with the *L. stagnalis* AChBPs, which is the model for the ligand-binding domains of the whole family of Cys-loop receptors [24,25,26]. Ln-CHOL was the most active (IC_50_ 2.3 µM) (Figure 2A, Table 1), whereas all three other compounds, as well as ACh, demonstrated activities only at five- to ten-fold higher concentrations (IC_50_~13 to 20 μM). 

Since acylcholines could be expected to exhibit detergent properties to the nAChRs in the cell membranes, we tested the effect of the widely used detergent sodium dodecyl sulfate (SDS) on the ^125^I-αBgt binding. In the presence of 40 μM SDS, the binding of ^125^I-αBgt to *Torpedo* membranes or GH4C1 cells did not differ from the control values (data not shown). After that, we studied the ability of acylcholines to compete with ^125^I-αBgt for binding to the full-size receptors, muscle-type nAChR from *T. californica* electric ray (Figure 2B) or human α7 nAChR (Figure 2C, Table 1). The Ln-CHOL, most efficient against the AChBP (having no membrane environment), was also most active against the Torpedo nAChR (IC_50_ 18.7 µM). At the same time, α7 nAChR was most efficiently inhibited by Ol-CHOL (IC_50_ 14.2 µM).

### 3.3. Acylcholines Inhibit nAChR Function

The interaction of acylcholines with the nAChR’s binding site for agonists and competitive antagonists shown by radioligand competition experiments could, in principle, induce receptor activation or, *vice versa*, inhibition of the activation due to competition with endogenous agonists acetylcholine and choline (in the case of α7 nAChR). To find out whether the acylcholines act as agonists or antagonists, we performed fluorimetric measurements of cytoplasmic calcium evoked by activation of nAChR. The ability of the compounds to induce calcium mobilization was studied in the SH-SY5Y cells endogenously expressing α7 nAChR. Neither of the three tested compounds (AA-CHOL, Ln-CHOL, and DHA-CHOL) induced a rise in the intracellular Ca^2+^, and thus showed no agonistic properties towards this receptor. To test their antagonistic activity against α7 nAChR, they were applied with ACh in the presence of PNU120596, a positive allosteric modulator of this homooligomeric nAChR subtype. All tested compounds at micromolar concentrations inhibited the acetylcholine-evoked Ca^2+^ rise in a concentration-dependent manner (Figure 3A). The IC_50_ values in this Ca^2+^ mobilization assay were in the range from 1 to 19 μM for AA-CHOL, Ol-CHOL, Ln-CHOL, and DHA-CHOL (see Figure 3A). Thus, no significant influence of the fatty acid residue structure on the activity was found.

AA-CHOL also inhibited mouse muscle nAChR expressed in *Xenopus laevis* oocytes (Figure 3B) with IC_50_ = 3.16 ± 0.26 μM. 

Previously, we have shown (including our paper [27]) that α-bungarotoxin interacts with GABAAR in a way similar to its interaction with the acetylcholine-binding protein or the ligand-binding domain of nicotinic receptors. Therefore, we expected that AA-CHOL could have a similar activity. However, it showed no functional inhibition of α1β3γ2 GABAAR (Appendix A) up to a concentration of 10 μM. The higher concentrations of the substance in this experiment settings were considered to have only nonspecific effects, if any, and thus were not tested.

### 3.4. In Vivo Muscle Relaxation Test

The observation of muscle nAChR inhibition by AA-CHOL with micromolar affinity led us to the hypothesis that acylcholines acts as muscle relaxants at high doses. A possible muscle relaxant effect of the AA-CHOL was studied by measuring the mouse front legs strength at test points of 15 min, 30 min, 1 h, 2 h, and 4 h, according to the protocol we previously used for azemiopsin, a peptide inhibitor of the muscle nAChR [28]. AA-CHOL was injected into the front paws in a volume of 20 μL for each paw in doses of 0.5, 1, and 4 mg/kg. The test group showed no differences from the control group (vehicle injection). We also tested AA-CHOL at higher doses using the subcutaneous injection of 10 and 30 mg/kg. No differences with the respective controls (vehicle) were found (Figure 4).

### 3.5. Acylcholines Inhibit the Viability of A549 Cells

Lung cancer cells endogenously express α7 nAChR, and their activation on these cells leads to stimulation of proliferation [29]. Because acylcholines demonstrated the ability to inhibit nAChR, we hypothesized that they could inhibit proliferation of a lung cancer cell line A549. Indeed, Ol-CHOL dose-dependently decreased A549 viability after a 24 h incubation, reaching 45% inhibition at the concentration of 100 μM. Ln-CHOL and AA-CHOL produced only a 10% decrease at the 100 μM concentration, while DHA-CHOL was inactive (Figure 5A). The treatment with Ol-CHOL was accompanied by phosphatidylserine exposure on the membrane, caspase 9 and caspase 3, but not caspase 8 activation (Figure 5B,C), thus, indicating apoptosis induction.

On the one hand, the specific blocker of nAChR methyllycaconitine did not affect A549 proliferation up to the concentration of 10 μM. Nicotine, on the other hand, slightly decreased A549 proliferation at the concentration 15 μM but did not affect the cytotoxicity of 90 or 100 μM of Ol-CHOL (Figure 5D).

### 3.6. Acylcholines Inhibit the Activity of Cholinesterases 

Because long-chain acylcholines are structurally similar to the acetylcholine, we hypothesized that the enzymes which metabolize the latter could also degrade the long-chain acylcholines. We tested the ability of purified AChE and BChE to hydrolyze AA-CHOL. ATCh, as ACh analogue for spectrophotometric Ellman assay was used as a control for the enzyme activity, and incubations with the heat-inactivated enzyme were used for non-enzymatic degradation controls. Under experimental conditions, ATCh was fully hydrolyzed by both enzymes. AA-CHOL was not hydrolyzed by AChE, but with BChE the hydrolysis of arachidonoylcholine accounted for up to 30% during 40 min. With the heat-inactivated enzymes, no hydrolysis of either ATCh or AA-CHOL was detected. 

To further check if acylcholines possess the ability to interact with these enzymes, we tested their inhibition of AChE and BChE catalyzed ATCh hydrolysis (Table 3). Only AA-CHOL was active in these experiments, although the obtained IC_50_ values were quite high.

The graphical analysis of steady-state inhibition kinetics data for AA-CHOL toward AChE and BChE is shown in Figure 6. Binding of AA-CHOL to AChE and BChE changed both *V*_max_ and *K*_m_ values. Such alterations are consistent with mixed-type inhibition (Figure 6). For AChE inhibition *K_i_*= 16.7 ± 1.5 μM and *αK_i_*= 51.4 ± 4.1 μM, for BChE inhibition *K_i_*= 70.5 ± 6.3 μM and *αK_i_*= 214 ± 17 μM. 

According to the molecular docking results, choline moiety of AA-CHOL was found in the AChE active site, binding to the cation-binding pocket, but carbonyl oxygen atom of the ester group was outside the oxyanion hole due to a large size of acyl chain, which makes hydrolysis reaction unlikely (Figure 7A). Arachidonic chain spans through the gorge and enters the AChE peripheral anionic site (PAS, separated by Y124 and Y341). In addition, docked poses of AA-CHOL were obtained in PAS only, which is consistent with mixed-type inhibition. BChE has a larger acyl-binding pocket and wider gorge, which binds the carbonyl oxygen atom of the ester group to the oxyanion hole. This makes the hydrolysis reaction possible (Figure 7B) and agrees with the experimental data. There were additional binding poses of AA-CHOL outside the active site, which was consistent with a mixed-type inhibition observed experimentally. Arachidonic chain was found bent in the lower part of the gorge; such difference in the binding mode of ligands with lengthy chains to AChE and BChE is quite typical due to the difference of the gorge radii [30,31]. 

## 4. Discussion

Cholines acylated with unsaturated fatty acids are a recently discovered family of endogenous lipids. While fatty acid derivatives of other neurotransmitters possess interesting neuro- and immunomodulatory properties, the data on the biological activity of acylcholines remain very limited. On the basis of the structure similarity, we hypothesized that the receptor targets and metabolizing enzymes for long-chain unsaturated acylcholines could be similar to those interacting with acetylcholine. To test this assumption, we studied four long-chain unsaturated acylcholines containing residues of arachidonic, oleic, linoleic, and docosahexaenoic acids in the nAChR binding, nAChR functional response and in the interaction with the ACh degrading enzymes. Indeed, we found that long-chain unsaturated acylcholines can interact with the ACh targets; in all cases, an inhibitory activity or receptor interaction was observed.

The acylcholines were chemically synthesized via the intermediate acyldimethylaminoethanols, as in [13]. The synthesis and purification procedure was quite straightforward, and the reaction yields were typical for the syntheses of the acylated neurotransmitters [32,33].

The first step was to test whether the long-chain unsaturated acylcholines interact with the acetylcholine-binding site of the nAChR. We chose the following three standard models for this task: *L. stagnalis* AChBP, *T. californica* muscle-type nAChR, and human neuronal α7 nAChR. The existing literature contains no data on long-chain acylcholine binding to AChRs, except for the data on allosteric binding of phosphocholine and phosphatidylcholine [34]. The efficiency of acylcholines association with the AChBP was almost the same as compared with the ACh binding, whereas, with the two nAChR subtypes, the efficiency was two to four times weaker. DHA-CHOL was virtually inactive in the competition experiments on the nAChRs, while more active ones were the less unsaturated acylcholines. A problem in the binding analysis of such molecules could be their possible detergent activity, as they possess both a hydrophobic tail and a charged head. However, a classical detergent sodium dodecyl sulfate at a concentration of up to 40 μM did not inhibit α-bungarotoxin binding in our setting, and so this activity could be ruled out. The fact that the extension of the acid residue significantly decreased binding to the nAChRs, but not to the AChBP (lacking the membrane environment), could be possibly explained by the sequestration of the acylcholines with the membrane lipids surrounding of the receptor. To test this, we calculated a measure for substances hydrophobicity, namely the logarithms of partition coefficient octanol-water using the ChemBioDraw 12 software. The obtained results (4.03, 4.51, 4.12, and 4.21 for Ln-CHOL, OL-CHOL, AA-CHOL, and DHA-CHOL, respectively) were in the range typical for other lipids that readily associate with membranes, e.g., sphingosine (logP 4.88), and thus agreed with this hypothesis.

To study the functional response to the acylcholines, we measured their ability to induce Ca^2+^ mobilization in the SH-SY5Y cells, which endogenously express α7 and some other nAChRs [35]. The effect on the α7 nAChR in these cells can be discerned only in the presence of PNU120596, a positive allosteric modulator (PAM) of this nAChR subtype. None of the tested compounds showed agonistic properties under these conditions. However, in the presence of PNU120596, the tested compounds with similar potency inhibited the acetylcholine-evoked Ca^2+^ rise in a concentration-dependent manner. We discovered such an inhibitory activity for the long-chain unsaturated acylcholines for the first time, while shorter acylcholines (like palmitoylcholine) are known to act as nAChR-like agonists in physiological tests [4]. The binding site in the nAChRs is located inside the cavity under the C-loop [36]. The specific geometric fit of a ligand to the site is needed to activate the receptor [37]. Fatty acid chain elongation could, in principle, disrupt the ligand fit to the site and convert an agonist to an antagonist. Such an effect could be the reason for the observed nAChR-blocking activity of long-chain unsaturated acylcholines.

A somewhat contradictory activity has been earlier described for AA-CHOL and DHA-CHOL on the unfertilized sea urchin eggs, which expressed receptors somewhat similar to the nAChRs [12]. In this model, AA-CHOL and DHA-CHOL dose-dependently induced the larva immobilization and cell lysis, and the noncompetitive nicotinic cholinergic antagonist QX-222 antagonized this effect [13]. Thus, AA-CHOL acted as a nAChR agonist or a cholinomimetic in this model. This discrepancy in the activity could be due to a possible significant difference between the mammalian and sea urchin AChRs. The latter could be more similar to the muscarinic ACh receptor (mAChR) [38], which has a different active site structure and an allosteric regulation site on the receptor surface [39].

Distinct nAChR subtypes are involved in the stimulation of cancer cell proliferation [40]. Because the tested acylcholines appeared to be the nAChR blockers, we analyzed their activity on the A549 lung cancer cells which endogenously express α7 nAChR [41]. Ol-CHOL, Ln-CHOL, and AA-CHOL, but not DHA-CHOL, dose-dependently decreased the A549 viability after a 24 h incubation, the first compound being the most active. The activity of acylcholines in this experimental setting correlated with the data on their receptor binding, and thus the action via ACh receptor seems highly probable. According to the literature data, the selective antagonist of nAChR methyllycaconitine at high concentrations (1 to 10 µM) inhibits the proliferation of the A549 cells. However, this effect is relatively small and manifests itself predominantly after 48 and 72 h of incubation [42]. Considering much lower affinity of acylcholines for nAChR, as compared with methyllycaconitine, their cytotoxicity might not arise solely from the nAChR inhibition. A plausible hypothesis could be the concomitant activation of mAChRs by acylcholines, as the blocking of M2 mAChR on A549 cells was shown to inhibit their proliferation [43], or a combined action via both receptor types.

With regards to the in vitro tests, the acylcholines behaved as nAChR blockers, and we supposed that they could also act as neuromuscular blocking myorelaxants in vivo. It is noteworthy that such activity was recently demonstrated for azemiopsin, a linear peptide from the snake venom inhibiting the muscle nAChR at micromolar concentrations [28,44]. However, AA-CHOL showed no difference from the control in mouse grip strength test after the intramuscular injection. There could be several explanations for this fact including: insufficiency of either dose or time, substance degradation by the local enzymes, and problems with substance diffusion. An insufficient dose or time could be ruled out, as there was no substance activity even after a 30 mg/kg injection and 240 min of observation. In comparison, azemiopsin produced a relaxant effect in the concentration range 30–300 µg/kg, 10 min after injection [28]. Substance degradation is also of low probability, as we did not observe acylcholines hydrolysis by AChE and only a very negligible degradation by BChE. Problems with substance diffusion are possible, as studied acylcholines are quite hydrophobic. However, AA-CHOL and its long-chain analogs tested in our experiments are endogenous compounds, and their even very low activity as muscle relaxants could be of physiological relevance. 

In addition to the action on the nAChR, we tested the acylcholines activity on another part of the acetylcholine signaling system, namely on the acetylcholine hydrolyzing enzymes. AChE did not hydrolyze the tested acylcholines, but AA-CHOL inhibited its activity as a mixed-type inhibitor with quite high inhibition constants, which agrees with molecular docking results. Neither Ln-CHOL nor DHA-CHOL had noticeable inhibitory activity. BChE was also rather weakly inhibited by AA-CHOL. Molecular docking confirmed the possibility of BChE catalyzed hydrolysis of AA-CHOL and mixed-type mechanism of its inhibition of ATCh hydrolysis. No data on the inhibition of the AChE and BChE by acylcholines could be found in the literature. The existence of other enzymes for degradation of acylcholines seems possible, but testing this hypothesis requires further studies.

Therefore, long-chain fatty acid derivatives of choline could be both inhibitors (via the receptor blocking) and weak potentiators (via the inhibition of acetylcholine hydrolysis) of the acetylcholine signaling. However, receptor inhibition appeared at concentrations in the range 10–50 μM, while enzyme inhibition required 100 to 300 μM of the substances. Thus, receptor blocking and acetylcholine signaling inhibition appear as a dominating activity of acylcholines. At present, there are no data on the concentrations and biosynthesis of these compounds, making the correct estimation of their biological role very difficult. The discovered cytotoxicity of Ol-CHOL against the lung cancer cell line A549 points to the existence of additional receptor targets for this substance, and clarification of this issue could be of interest.

## 5. Conclusion

For the first time, we have demonstrated that arachidonoylcholine and its unsaturated fatty acid analogs with the chain length of 18 and 22 carbon atoms are inhibitors of the neuronal and muscle-type nicotinic acetylcholine receptors and modest inhibitors of AChE and BChE, thus, playing a possible role as endogenous modulators of the acetylcholine signaling system.

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
