# Peer review of "Arachidonoylcholine and Other Unsaturated Long-Chain Acylcholines Are Endogenous Modulators of the Acetylcholine Signaling System"

_biomolecules, 2020, doi:10.3390/biom10020283_

Round 1

Reviewer 1 Report

The study of Bezuglov et al deals with the impact of acylcholines as modulators of the acetylcholine signaling pathway. Importantly acylcholines have been found in many tissues and recent papers have shown that in particular unsaturated long-chain acylcholines are altered in disease, such as cardiovascular diseases. The authors focus on these newly identified metabolites, arachidonic (AA) linoleic (Ln), Oleic (Ol) and DHA-Cholines. Therefore, the results are interesting for a broad readership, where alterations of these metabolites have been found, as well as scientists dealing with lipids in general.  In my opinion, the paper is in the clear focus of the journal and the presented data are new and relevant.  I have a few recommendations and comments before this paper is suitable for publication.

General/major  comments:

For someone not dealing with lipids and acylcholines the rationale and the aim of the experiments are in parts not easy to understand. I would recommend to work and point out the aim, the key findings and especially the biological relevance more precisely in the abstract. Moreover, the important in vivo mouse data is not even mentioned there.

In line with this, many different cell lines were used; what was the rationale for using exactly these cells.

Please go carefully through the material and methods section, e.g. some cell lines, substances, etc are not mentioned.

In some cases, it has been mentioned that some experiments are performed two or three independent times (e.g. table 1). Please make sure that the results were obtained from at least 3 independent experiments to provide a proper statistic.

Please make sure that the figure/ table legends fit to the data. E.g. table 1 SEM is mentioned also no SEM is provided

Please add data not shown in line 272 to the supplement.

Why have the authors restricted the upper concentration range to 10 µM.

Figure 2a. How have the curves been fitted to the data? E.g. for Ln-Chol other curve progressions seems to be possible.

Please provide a statistic if there is a significant difference between the different acylcholines.

Unfortunately, the authors did not investigate the effects of EPA-cholines. A recent study points out a strong link to Vitamin D metabolism and osteoporosis. I would appreciate that EPA-Cholines would be integrated in this study as well.

Please add the quality check of the synthesized acylcholines at least in the supplement.

Please perform a caspase assay in addition to the MTT assay to check for apoptosis

Minor comments:

Please chose more adequate keywords. SH-SY5Y as a key word is very unspecific for the topic

Author Response

Point 1. For someone not dealing with lipids and acylcholines the rationale and the aim of the experiments are in parts not easy to understand. I would recommend to work and point out the aim, the key findings and especially the biological relevance more precisely in the abstract. Moreover, the important in vivo mouse data is not even mentioned there.

Response 1. The abstract was reformulated to contain the work’s hypothesis, key findings, and biological relevance. However, the abstract length is limited to only 200 words, and thus we decided to spare the in vivo data on the absence of AA-CHOL activity for the data on those tests in which the activity was present.

Point 2. In line with this, many different cell lines were used; what was the rationale for using exactly these cells.

Response 2. In this work, four cell models were used for different purposes. 1) Xenopus leavis oocytes is a standard electrophysiological model; 2) commercially available GH4C1 cells membranes overexpressing α7 neuronal nAChR provide a required receptor density for receptor radioligand binding studies; 3) SH-SY5Y cells were used as a convenient calcium imaging model with a known endogenous expression of α7 nAChR; 4) A549 lung cancer cells were used as a clinically related model of nAChR function, and if acylcholines had demonstrated a significant activity in this model, these data could have clinical significance as well.

Point 3. Please go carefully through the material and methods section, e.g. some cell lines, substances, etc are not mentioned.

Response 3. The methods section was extended with the sources of T. californica membranes and L. stagnalis AChBP; all the used reagents sources were indicated

Point 4. In some cases, it has been mentioned that some experiments are performed two or three independent times (e.g. table 1). Please make sure that the results were obtained from at least 3 independent experiments to provide a proper statistic.

Response 4. All experiments were performed at least in triplicate

Point 5. Please make sure that the figure/ table legends fit to the data. E.g. table 1 SEM is mentioned also no SEM is provided

Response 5. The legends were checked for consistency. In the legend for Table 1, the incorrect mention of SEM was removed (the values are presented with 95% CIs). In the legend of Figure 1, the SEM mention was added

Point 6. Please add data not shown in line 272 to the supplement.

Response 6. The data on GABAAR electrophysiological study were added to the supplement (Figure S5)

Point 7. Why have the authors restricted the upper concentration range to 10 µM.

Response 7. The higher concentrations of the substance in GABAAR electrophysiology experiment settings were considered to have only non-specific effects, if any, and thus were not tested.

Point 8. Figure 2a. How have the curves been fitted to the data? E.g. for Ln-Chol other curve progressions seems to be possible.

Response 8. The curve fitting algorithm is described in the Materials and methods section. The radioligand assay results were analyzed with ORIGIN 8.0 (OriginLab Corporation, Northampton, MA, USA) fitting to a one-site dose-response curve by the equation: % response = 100/{1 + ([toxin]/IC50)p}, where IC50 is the concentration at which 50% of the binding sites are inhibited, and p is the Hill coefficient.

Point 9. Please provide a statistic if there is a significant difference between the different acylcholines.

Response 9. All tables legends and Figure 5 legend were supplemented with the ANOVA results

Point 10. Unfortunately, the authors did not investigate the effects of EPA-cholines. A recent study points out a strong link to Vitamin D metabolism and osteoporosis. I would appreciate that EPA-Cholines would be integrated in this study as well.

Response 10. Thank you very much for the good point. The study of EPA-CHOL activity, indeed, looks interesting. However, in most of our models we did not observe a substantial influence of the fatty acid residue on the acylcholine activity, and thus we expect EPA-CHOL to behave very similar to the substances already present in this study. In addition, the main topic of this paper was long chain acylcholines activity within the nicotinic acetylcholine signaling system, and not osteoporosis. Therefore, the inclusion of EPA-CHOL in this study does not seem to be necessary, but we will try to devise an osteoporosis model in the future and study EPA-CHOL activity within it.

Point 11. Please add the quality check of the synthesized acylcholines at least in the supplement.

Response 11. Microcolumn HPLC data for the compound purity were added to the supplement (Supplement figures 1-4)

Point 12. Please perform a caspase assay in addition to the MTT assay to check for apoptosis

Response 12. To check for the apoptosis induction, caspase 3, 9, and 8 were assayed using specific fluorogenic substrates. In addition, the cells were stained with the apoptosis/necrosis detection kit

Point 13. Please chose more adequate keywords. SH-SY5Y as a key word is very unspecific for the topic

Response 13. The keyword SH-SY5Y was removed

Reviewer 2 Report

The manuscript by Akimov and co-workers looks like a nice contribution for the  understanding of the interactions of acylcholines with the acetylcholine signaling system. The submitted version of the manuscript is well done but presents some issues that should be addressed before publication, as commented below:

1) The word "etc" is not appropriate for a scientific publication. Please revise the text and remove it;

2) Respect the abbreviations. Once an abbreviation is defined it should be used throughout the text instead of the original word. AChE and BChE, for example, were observed to be abbreviated several times in the text. Just once is enough. Please revise the whole text for other cases;

3) Introduction brings some typos, grammar issues and phrases out of context. Please revise it carefully and refine the text;

4) Scheme 1 is not a scheme. It's rather a figure;

5) Please include the reference [13] in the legend of Scheme 2.

Author Response

Point 1.  The word "etc" is not appropriate for a scientific publication. Please revise the text and remove it;

Response 1. The phrase with the word ‘etc’ was reformulated.

Point 2.  Respect the abbreviations. Once an abbreviation is defined it should be used throughout the text instead of the original word. AChE and BChE, for example, were observed to be abbreviated several times in the text. Just once is enough. Please revise the whole text for other cases;

Response 2. Text revision was done to make the abbreviations use consistent.

Point 3. Introduction brings some typos, grammar issues and phrases out of context. Please revise it carefully and refine the text;

Response 3. The errors in the introduction were corrected

Point 4. Scheme 1 is not a scheme. It's rather a figure;

Response 4. Scheme 1 was renamed to Figure 1.

Point 5. Please include the reference [13] in the legend of Scheme 2.

Response 5. The reference [13] was included in the legend of Scheme 2

Reviewer 3 Report

Dear Editor,

I am sending you my referee report on the manuscript entitled: Arachidonoylcholine and other unsaturated long-chain acylcholines are endogenous modulators of the acetylcholine signaling system, which was recently submitted to the journal -  Biomolecules (Ms No biomolecules-702213).

According to my opinion it is an interesting original article, which is aimed at long-chain unsaturated acylcholines as endogenous modulators of the acetylcholine signaling system.

It is a very complex study.

I have just one remark regarding AChE and BuCHe inhibition.

Are authors really sure, that they found real inhibition of cholinesterases.

I did similar experiments decade before with other long chain molecules.

And I found, that there is other type of enzyme inhibition.

As these compounds have long alkylating tail, they behave as detergents.

Due to this, they are destroying enzyme. And that is a reason why enzyme is not working at higher concentrations - and it looks like inhibition.

Authors should consider this - and perhaps omit the inhibition of AChE section.

Or they should give real proof, that they inhibit the enzyme, and which type of inhibition was found (competitive, uncompetitive, etc.)

Finally, just MINOR REVISIONS NEEDED

Author Response

Point 1. I have just one remark regarding AChE and BuCHe inhibition. Are authors really sure, that they found real inhibition of cholinesterases. I did similar experiments decade before with other long chain molecules. And I found, that there is other type of enzyme inhibition. As these compounds have long alkylating tail, they behave as detergents. Due to this, they are destroying enzyme. And that is a reason why enzyme is not working at higher concentrations - and it looks like inhibition. Authors should consider this - and perhaps omit the inhibition of AChE section. Or they should give real proof, that they inhibit the enzyme, and which type of inhibition was found (competitive, uncompetitive, etc.

Response 1.  According to reviewer’s remark regarding AChE and BChE inhibition, we performed a detailed kinetic study of AChE and BChE inhibition by AA-Chol, which demonstrated mixed-type inhibition.  We also studied in more detail the inhibition of both enzymes by AA-CHOL, increasing the number of repetitions to 17, which allowed us to significantly refine the inhibition constants. Results are presented in the manuscript. Also we performed molecular docking of AA-Chol into AChE and BChE active site gorge, which are in accord with the kinetic experimental results.

Round 2

Reviewer 1 Report

The authors have adressed all my comments and concerns sufficiently. The manuscript is clear and not speculative and the conclusions are supported by the data. Therefore I recommend the manuscript for publication in its present form.